# Impulsivity and Compulsivity and Their Relationship with Non-Adherence to Treatment in the Prison Population

**DOI:** 10.3390/ijerph18168300

**Published:** 2021-08-05

**Authors:** Francisca López-Torrecillas, Eva Castillo-Fernández, Isabel Ramírez-Uclés, Ignacio Martín

**Affiliations:** 1Center for Research into the Mind, Brain, and Behavior (CIMCYC), Faculty of Psychology, University of Granada, 18071 Granada, Spain; 2Albolote Penitentiary Center Granada, 18220 Granada, Spain; evacastillofer@correo.ugr.es; 3Department of Personality, Assessment and Psychological Treatment, The National Distance Education University, 28040 Madrid, Spain; iramirez@psi.uned.es; 4Department of Methodology for the Behavioral Sciences, Faculty of Psychology, University of Granada, 18071 Granada, Spain; imartin@ugr.es

**Keywords:** impulsivity, compulsivity, treatment adherence, prison population

## Abstract

The main challenge of interventions in penitentiary institutions is the re-education and reintegration of inmates, i.e., that inmates have the intention and ability to live law-abiding lives. Therefore, an increase in self-control or, on the contrary, the decrease or elimination of impulsive versus compulsive behaviors becomes necessary. This study aimed to evaluate the role of impulsivity versus compulsivity in treatment adherence in the prison population. The study included 134 men from the Penitentiary Center of Granada who were divided into two groups. Group 1 was considered treatment adherent, and Group 2 was considered non-adherent to treatment. The following instruments were used: Symptom Inventory (SCL-90-R), Addiction Severity Index (EuropASI), Impulsivity Scale (UPPS-P), and Compulsive Belief Questionnaire (OBQ-44). Statistically significant differences were found in impulsivity in the dimensions of negative urgency, sensation seeking, and positive urgency, with higher scores in all cases for the non-adherent group than for the adherent group. We also found statistically significant differences in responsibility/inhibition, perfectionism/uncertainty, and importance/control, with higher scores for the non-adherent group compared to the adherent group. Treatment adherence is inversely related to impulsive and compulsive behaviors.

## 1. Introduction

Treatment adherence is defined as the degree to which the person’s behavior follows the prescribed recommendations of the therapist. It includes the patient’s ability to attend scheduled appointments, take prescribed medications, make recommended lifestyle changes, and complete self-reports, tests, or homework assignments as requested [1,2]. There is currently considerable consensus [3] that physical activity and healthy eating habits (consumption of fruits and vegetables) are fundamental behaviors in disease prevention and indicate adherence to treatment. Along the same lines, a meta-analysis [4] analyzing adherence in hospital and community interventions found that healthy eating habits, community support, group intervention, and knowledge of the symptomatology related to the disease being suffered are crucial elements in supporting adherence to treatment. Moreover, individual factors (such as elderly age and treatment utility beliefs) impact treatment non-adherence and affect treatment success in both chronic and acute diseases [5]. It has also been found [1,6] that non-adherence and noncompliance with homework are related to personal variables, such as impulsivity. Impulsivity is associated with various risky activities (potentially escalating to criminal acts) and poor treatment outcomes [7].

The American Psychiatric Association [8] defines impulsivity as a predisposition to undertake rapid and unplanned reactions to internal or external stimuli without considering the negative consequences. Impulsivity has been consistently linked to a lack of behavioral inhibition and is responsible for risky behaviors, such as drug use and aggression [9]. In addition, behavioral inhibition is considered a fundamental component of executive function. Executive function is determined by complex cognitive processes involved in thought control and mediated by the frontal lobe, more specifically, the prefrontal cortex [10].

Aggressive behavior has been associated with two distinct subtypes: impulsive and premeditated (compulsive) [11]. Impulsive aggression is defined as an aggressive response triggered by a provocation leading to loss of behavioral control, while premeditated (compulsive) aggression is defined as an intentional or conscious aggressive act that is not spontaneous or related to a state of agitation due to anger problems. The link between aggression and impulsive or compulsive disorders has been associated with inefficient frontal lobe function and are terms that reflect complex neurocircuitry [12].

Impulsivity and compulsivity are thought to underlie violent behaviors. Impulsivity is related to decreased cognitive control, increased risk-taking, and behavioral disinhibition (e.g., drug addiction). Compulsivity is associated with an excess of behavioral control that can lead to an initial reluctance to engage in a particular behavior and ultimately promote repetitive behaviors and avoidance of harm, e.g., violent acts might be carried out without warning (impulsive act) or after a great deal of rumination (compulsive act) [13]. Recent advances in understanding the neurocircuitry of impulsivity and compulsivity have led to the idea that many maladaptive behaviors, such as the commission of crimes, share these two dimensions (impulsivity and compulsivity). Impulsive and compulsive disorders, including aggression and thus the commission of crimes, are caused in part by inefficient information processing in executive functions [12]. Impulsive and compulsive behaviors are both characterized by the inability to inhibit or delay behavior, which is related to the prefrontal cortex. Compulsive behavior appears to be associated with increased frontal lobe activity, whereas impulsive behavior may be associated with reduced activity in this region [14]. The functions of the prefrontal cortex are to inhibit specific behaviors or actions, i.e., to override impulsive and compulsive tendencies and to make decisions. Therefore, damage to the prefrontal cortex may interfere with the functioning of these processes (impulsivity and compulsivity). To date, no studies have analyzed the role of compulsivity in treatment adherence. Thus, the present study aimed to evaluate the role of impulsivity versus compulsivity in treatment adherence in the prison population.

To achieve the objectives of this study, we first proceeded to verify whether the impulsivity variable assessed through the Impulsivity Scale (UPPS-P [15]) differed between the groups. Second, we checked whether compulsivity assessed through the self-report Obsessive-Compulsive Belief Questionnaire (OBQ-44; [16]) also differed between the groups (adherent versus non-adherent).

## 2. Materials and Methods

### 2.1. Study Design and Setting

The present study employed a cross-sectional observational design using standardized assessment instruments adapted to the Spanish population. A total of 134 males from the Granada Penitentiary Center with a mean age of 37.48 years (SD = 8.31; 20–54 years) participated in this study. Participants were receiving a reintegration intervention implemented by the prison team. Cognitive-behavioral and psychoeducational therapy were included as part of this treatment. The participants were divided into two groups based on criteria of the specialized prison technical staff and considering the recommendations of the reviewed literature on treatment adherence [1,17]. Group 1 was considered treatment adherent and comprised 56 men with a mean age of 38.7 years (SD = 7.6). Group 2 was considered non-adherent and consisted of 78 men with a mean age of 36.6 years (SD = 8.7). The inclusion criteria were serving a sentence in the said penitentiary center and voluntarily participating in the study. Exclusion criteria were being over 55 years, having a physical or psychiatric illness (schizophrenia and/or depression), and receiving current psychopharmacological treatment for the diagnosed illness. To assess the exclusion criteria, participants completed the Symptom CheckList (SCL-90-R; [18]). Participants were informed about the aims of the study and provided signed informed consent to participate anonymously.

The socio-demographic data and those related to drug use and crimes were handled by the prison team that collaborated in this study with the patients’ consent. Table 1 displays the socio-demographic characteristics of the participants as well as those related to drug use, crimes, and convictions.

### 2.2. Procedure and Instruments Used

First, participants were interviewed individually to check the inclusion criteria and to confirm their participation in the research. Participants were reminded at the beginning of the session of their right to discontinue the procedure at any time, and their written consent was then obtained. The participants filled the measurements of this study as pen-and paper by the participants themselves. This process was supervised by psychologists. All patients were informed about the study course and methods, and about the possibility of withdrawing from the study at any time. All patients provided their written informed consent to participate in the anonymous survey.

The participants were receiving intervention for their reintegration, implemented by the prison team. This is a program designed to modify criminal behavior for positive development within the prison in the short term and reinsertion and rehabilitation in the long term. Prison treatment is regulated in Section III of the 1979, General Penitentiary Organic Law (LOGP), Articles 59 to 72, and in Section V of the 96 Regulations, Articles 110 to 153. According to Article 59 of the LOGP [19], prison treatment consists of a series of activities aimed at achieving re-education and reinsertion. The goal of the treatment is to ensure that the inmate has the intention and ability to live in a manner that respects the criminal law, and to advise on their needs. Therefore, it covers four fundamental areas: regulatory, well-being, training-cultural-employment, and the therapeutic area [20]. Within the therapeutic area, one of the priority objectives in the prison setting is to enable inmates to live without drugs. As part of the treatment, participants received cognitive-behavioral and psychoeducational therapy. The program combines individualized, or group actions developed by the prison psychologist to manage emotions, cognitions, and behaviors that impede the development of adequate self-control on the part of the inmate.

Treatment adherence was determined through the reports recorded by the psychologist of the module to which the participants belonged. These reports had been prepared in accordance with the published proposals [1,17] on the key elements to be considered when preparing a treatment adherence report. These elements were: Compliance with out-of-session activities after the intervention session (known as homework); completion of self-reports; maintenance of abstinence from alcohol and drug use (tobacco was not included); attendance at therapeutic sessions, attendance at workshops; participation in leisure activities programmed by the prison in the sociocultural center (e.g., cinemas, concerts, and conferences); daily physical activity; maintenance of personal cleanliness, personal area (cell), and common areas; eating the food prepared in the prison center (because it is balanced and healthy) and avoiding buying and eating junk food from the commissary. This information was transformed into a dichotomous variable (adherence, non-adherence).

The measurements used in this study are described below:

Report on demographic variables, crimes, and behaviors in institutions. This report was completed by the prison psychologist who was responsible for collecting socio-demographic data, along with information on the types of offenses and penalties, and sanctions within the prison following the prison regulations [21].

The Symptom Checklist (SCL-90-R; [18]). This is a symptom scale that evaluates the degree of psychological distress experienced by a person for a period of one week prior to the time of assessment. It consists of 90 items (a reduced version of 52) with a Likert-type scale composed of five response options. This instrument is structured into nine primary dimensions (somatizations—SOM, obsessions and compulsions—OBS, interpersonal sensitivity—IS, depression—DEP, anxiety—ANS, hostility—HOS, phobic anxiety—FOB, paranoid ideation—PAR, and psychoticism—PSIC), seven additional items (referring to sleep disorders, eating behavior, thoughts about death and feelings of guilt), and three global distress indices (Global Severity Index—GSI, is an indicator of the current level of perceived distress, total positive symptoms—TSP refers to the total number of symptoms recognized as present, and the Positive Symptomatic Distress Index—PSDI assesses response style). Reliability analyses reveal that the nine dimensions yielded values close to or higher than 0.70. The scale has high validity (concurrent and predictive), using as criteria other clinical assessment instruments, screening scales, psychiatric diagnoses, structured assessment protocols, or relapse indicators. This is published in TEA editions [22].

Addiction Severity Index, European version (EuropASI; [23]). This is a semi-structured and standardized interview that assesses the severity of alcohol and drug dependence by analyzing problems related to consumption. It provides information on the patient’s situation at the time of assessment. It consists of 150 items grouped into six scales: (1) medical situation (16 items); (2) work and social situation (26 items); (3) drug and alcohol consumption (28 items); (4) legal situation (23 items); (5) family and social relationships (26 items), and (6) psychopathological state (22 items). In each area, objective questions are asked that measure the number, extent, and duration of problem symptoms throughout the patient’s life and in the last 30 days. Its application lasts between 40 and 50 min. Information is obtained on the severity of the problem, which is quantified on a scale from 0 to 9. It is a hetero-rated scale, and prior training is necessary. After completing each scale, the participant is asked about their degree of concern or discomfort. These subjective items are rated by the patient on a 5-point scale: 0 (not at all), 1 (slightly), 2 (moderately), 3 (considerably), and 4 (extremely). This provides the interviewer with an assessment of the truthfulness of the information reported by the patient, specifically regarding the intention to provide distorted responses. The interviewer’s assessment is based on the responses given to the objective items. It provides a rating estimated by the evaluator with a score ranging between 0 and 9. Concerning the validity of the interviews that assess the severity of the disorder, it should be noted that, from the clinical point of view, EuroPASI provides information to describe the needs of patients at the beginning of treatment, assign patients to appropriate therapeutic strategies, and evaluate the results of interventions. Both ASI and EuropASI are widely used interviews whose reliability and validity have been confirmed by recent research [24,25,26].

Impulsivity Scale (UPPS-P; [15]). This is composed of 59 items measuring five dimensions of impulsivity: negative urgency (12 items), lack of perseverance (10 items), lack of premeditation (11 items), sensation seeking (12 items), and positive urgency (14 items). The items are scored on a Likert scale ranging from 1 (strongly agree) to 4 (strongly disagree). The Spanish version of the scale has shown good psychometric properties [27].

Obsessive Beliefs Questionnaire (*OBQ-44;* [16]). This evaluates dysfunctional (obsessive) beliefs. It consists of 44 Likert-scale items ranging from 1 (strongly disagree) to 7 (strongly agree). It is composed of three dimensions, which are responsibility and overestimation of threat (OBQ-RT) with 16 items, perfectionism and intolerance of uncertainty (OBQ-PC) with 16 items, and importance and control of thoughts (OBQ- ICT) with 12 items. The questionnaire presents adequate levels of reliability, an internal consistency of 0.95 for the total score, 0.89 for the responsibility/estimation of threat dimension, 0.88 for the perfectionism/uncertainty dimension, and 0.85 for the importance/control of thoughts dimension. With respect to the test-retest reliability, the coefficients are high for both the total (0.80) and for the dimensions (responsibility/estimation of threat = 0.74; perfectionism/uncertainty = 0.75; importance/control of thoughts = 0.79) [28].

### 2.3. Ethical Considerations

The Ethics Committee of the University of Granada approved the study (approval number 396/CEIH/2018). Participation was voluntary and anonymous, and all participants were informed about the purpose, methods, and course of the study and their right to refuse or discontinue participation. Written informed consent was obtained from each participant before inclusion, and the research complied with the principles proposed by the Declaration of Helsinki [29].

### 2.4. Statistical Methods

The variables (socio-demographic variables, such as marital status, educational level, crimes committed (which could be up to two), and history of drug use and severity of use) were compared using contingency tables and the likelihood-ratio test (Chis-Square—χ2) was applied. Second, analysis of variance (ANOVA) was conducted using a between-groups factorial design with group (adherent versus non-adherent) as the independent variable, and the age of the participants as the dependent variable. Third, multivariate analysis of variance (MANOVA) was performed with an intergroup factorial design using group (adherent versus non-adherent) as the independent variable and the scores on the six dimensions that comprise the Drug Use Severity Index (EuroPASI—medical situation; work and social situation; drug and alcohol use; legal situation; family and social relationships, and psychopathological state) as the dependent variables. Fourth, in order to achieve the objectives of this study, two multivariate analysis of variance (MANCOVA) were carried out with a one-factor between-groups design, using, in the first case, group (adherent versus non-adherent) and the variables of the Impulsivity Scale (UPPS-P: Lack of premeditation; negative urgency; lack of perseverance; sensation seeking; positive urgency) as independent variables and the EuroPASI total score as dependent variables and as covariate variables. Univariate ANCOVAs were then conducted for each of the dependent variable levels (lack of premeditation; negative urgency; lack of perseverance; sensation seeking; positive urgency) using group as the independent variable and the EuroPASI total score as a covariate. In the second multivariate analysis of variance (MANCOVA), with a one-factor between-groups design, the independent variable was group (adherent versus non-adherent) and the variables of the Obsessive Beliefs Questionnaire (OBQ-44) (responsibility/inhibition; perfectionism/uncertainty; importance/control) were dependent variables, while the EuroPASI Total Score was used as a covariate. Subsequently, univariate ANCOVAs were conducted for each of the dependent variable levels (responsibility/inhibition; perfectionism/uncertainty, and importance/control) using group as an independent variable and the EuroPASI Total Score as a covariate.

## 3. Results

### 3.1. Socio-Demographic, Delinquency, and Drug Use Variables of the Sample Studied

The study included 134 male participants from the Albolote Penitentiary Center of Granada with a mean age of 37.48 years (SD = 8.31; 20–54 years). The participants were divided into two groups according to their adherence to treatment. Group 1 was considered treatment adherent and was composed of 56 men with a mean age of 38.7 years (SD = 7.6). Group 2 was considered non-adherent and was composed of 78 men with a mean age of 36.6 years (SD = 8.7). The groups were age matched (F_1,132_ = 2.211; Mce = 151.412; *p* = 0.139). Table 1 shows the results obtained for the socio-demographic, delinquency, and drug use variables of the sample.

### 3.2. Results of Impulsivity Between-Group Comparisons (Adherent versus Non-Adherent)

The results derived from Impulsivity Scale (UPPS-P; [15]) revealed a statistically significant main effect of group (Wilks’ Lambda = 0.741, F_5,127_ = 8.858; *p* < 0.001). The results of the ANCOVAs showed that there were statistically significant differences in negative urgency (F_2,131_ = 16.437; measure of the effect size (Mez)— = 688.238; *p* < 0.001), sensation Seeking (F_2_,_131_ = 9.593; Mez = 552.028; *p* < 0.001), and positive urgency (F_2_,_131_ = 10.627; Mez = 905.213; *p* < 0.001), with the non-adherent group obtaining higher sores than the adherent group in all cases. No statistically significant differences were found in the variables of lack of premeditation and lack of perseverance. Means, standard deviations, significance, and effect sizes are shown in Table 2.

### 3.3. Results of Obsessive Belief Between-Group (Adherent versus Non-Adherent) Comparisons

The results derived from Obsessive Belief Questionnaire (OBQ-44; [16]) revealed statistically significant group differences in responsibility/inhibition (F_2,131_ = 4.761, Mez = 1202.233; *p* < 0.05), perfectionism/uncertainty (F_2_,_131_ = 5.317, Mez = 1473.834; *p* < 0.01), and importance/control of thoughts (F_2_,_131_ = 4.138, Mez = 763.739; *p* < 0.05), with the non-adherent group obtaining higher scores than the adherent group. Means, standard deviations, significance, and effect sizes are displayed in Table 3.

## 4. Discussion

Impulsivity and compulsivity underlie violent behaviors. Impulsive behavior appears to be associated with reduced frontal lobe activity, whereas compulsive behavior appears to be associated with increased activity in this region. Treatment adherence studies have shown that impulsivity plays a negative role, whereas nothing is known about the role of compulsivity in treatment adherence. This study therefore aimed to assess the role of compulsivity and impulsivity in treatment adherence in the prison population.

The objective of treatment in penitentiary institutions is the re-education and reintegration of inmates, i.e., to ensure that the inmate has the intention and the ability to live law-abiding lives. Therefore, when looking at adherence to this type of intervention, the role of self-control in particular, and of executive functions in general, are elements that cannot be overlooked. The ability to plan, organize, guide, review, regulate, and evaluate behavior is necessary for the inmate to adapt effectively to the prison environment and achieve both short- and long-term goals. To do this, the inmate will need to re-organize his/her belief system, motives, and values. Similarly, therapy in correctional institutions should ensure adequate intervention for both increased frontal lobe activity (which characterizes impulsive behavior) and decreased frontal lobe activity (which characterizes compulsive behavior).

Our study aimed to evaluate the role of impulsivity and compulsivity in treatment adherence in the prison population.

In this study, we found statistically significant differences in impulsivity in the dimensions of negative urgency, sensation seeking, and positive urgency, with the non-adherent group obtaining higher scores than the adherent group. Negative urgency is understood as the tendency to engage in risky behaviors or act rashly under negative affectivity, regardless of the negative consequences that could ensue. Positive urgency is understood as the tendency to engage in risky actions or lose control when faced with intense positive emotions. Moreover, sensation seeking is defined as the tendency to engage in and enjoy activities with a high emotional component and to be open to new experiences that could be dangerous. This study suggests that to increase treatment adherence, we should implement intervention programs aimed at managing intense emotional states (both positive and negative) to regulate impulsive behavior. In addition, in such programs, we should also encourage activities of daily living that increase the degree of personal well-being so that inmates do not have to resort to risky behaviors that negatively affect health, the legal system, and society. These results are consistent with previous studies [6,7] in which smokers with high impulsivity scores were found to show non-adherence to treatment. Likewise, our results are congruent with those obtained by another study [20] in which it was found that most inmates in correctional facilities presented characteristics related to impulsivity.

We also found statistically significant differences in responsibility/inhibition, perfectionism/uncertainty, and importance/control, with the non-adherent group again showing higher scores compared to the adherent group. These results suggest that non-adherent participants have higher scores on dysfunctional obsessive-compulsive beliefs compared to adherent participants. Dysfunctional obsessive-compulsive beliefs include an inflated sense of responsibility, overestimation of threat, perfectionism, intolerance of uncertainty, the over importance of thoughts, and the need to control such thoughts. We can consider our results to be entirely novel since, to date, no study has been conducted that analyzes the role of compulsivity in treatment adherence in the prison population. These results suggest that to increase adherence to treatment, we should design intervention strategies that address conceptual rigidity, overestimation of threats, excessive responsibility, and perfectionism. The first two (conceptual rigidity and overestimation of threat) are characteristics that make it challenging to design an intervention adapted to the prison context, since this is a rigid and very strict environment in terms of compliance with the rules.

However, the latter two elements (excessive responsibility and perfectionism) could be more readily addressed within the penitentiary context and could even favor the acquisition of penitentiary benefits (e.g., obtaining permission to leave, or gaining the social approval of the professionals of the penitentiary center). However, we have found that these characteristics (excessive responsibility and perfectionism) are related to non-adherence to treatment, whereas it might be logical to think that these traits are related to adherence to treatment. Both of these traits can be difficult to manage due to constant overexertion and dissatisfaction with the fulfillment of tasks; hence, after starting treatment, they have problems with adherence. When things do not go as they wish, behaviors incompatible with adherence to treatment emerge [30].

In summary, the results of this research lead to clinical applications in four ways: firstly, to design adequate frontal lobe dysfunction intervention; secondly, to encourage activities necessary for independent living at the prison or in the community; thirdly, to plan coping strategies that approach new challenges, tasks, and problems each day; and finally, fourthly, to the performance of daily activities that achieve the proposed goals.

This study is the first investigation to analyze the role of impulsivity and compulsivity concerning adherence to treatment in the inmate population. However, this work is not without limitations. For example, we consider the measurement instrument used to assess compulsivity to be a significant limitation, although this is currently the only available instrument that has been adapted and validated for the Spanish population to assess impulsivity. A second limitation is that the study sample was composed only of men. This is because we considered the crime of gender violence, which is defined as aggression by men against women, and because the male prison population is five times larger than that of women so that the selection of a gender-balanced sample (based on the inclusion and exclusion criteria) would not have been possible.

## 5. Conclusions

Treatment adherence is related to the management of intense emotional states (both positive and negative); the performance of daily activities that generate well-being or reduce threats; the ability to adapt to novel, rigid, changing, or unexpected situations; and the achievement of the proposed goals.

## Figures and Tables

**Table 1 ijerph-18-08300-t001:** Socio-demographic, delinquency, and drug use variables.

Variables	Group Adherent	Group Non-Adherent	F	*p*
**Marital Status** **(*N*)**			4.902	0.298
Single	31	31		
Married	9	13		
Divorced	7	10		
Widower	0	1		
Cohabiting with a partner	9	23		
**Educational Level** **(*N*)**			3.302	0.347
No Primary	10	13		
Primary	21	37		
Secondary/High School	25	26		
Undergraduate/Bachelor’s Degree	0	2		
**Type of Crime 1** **(*N*)**			13.630	0.009
Against life and integrity	4	7		
Against Freedom	2	4		
Against Property/ Treasury	29	47		
Against Public Health	11	1		
Gender Violence	10	19		
**Type of Crime 2** **(*N*)**			9.112	0.105
No crime	10	6		
Against life and integrity	6	16		
Against Freedom	1	2		
Against Property; Public Treasury	28	42		
Against Public Health	8	12		
Gender Violence	3	0		
	MEAN (SD)	MEAN (SD)	F	*p*
**Time of Sentence Served**	89.71 (71.53)	103 (91.57)	0.819	0.367
**Time of Sentence Served in Prison**	39.73 (42.39)	54.26 (49.27)	3.176	0.077
**EuropASI**	MED (DT)	MED (DT)	F	*p*
Physical/health	3.02 (2.15)	3.63 (2.35)	2.353	0.127
Employment	7.91 (1.77)	8.40 (1.28)	3.406	0.067
Alcohol	3.63 (2.20)	4.09 (2.12)	1.515	0.221
Drugs	4.89 (2.20)	4.82 (2.06)	0.038	0.846
Legal	9.00 (0.00)	9.00 (0.00)		
Family	3.52 (2.75)	3.82 (2.39)	0.460	0.499
Psychological	7.46 (2.84)	8.23 (2.06)	3.292	0.072
Total Score	39.43 (6.08)	41.99 (5.62)	6.303	0.013

Addiction Severity Index—Addiction Severity Index.

**Table 2 ijerph-18-08300-t002:** Mean, standard deviation, and significance level of differences in impulsivity (UPPS-P) between the groups.

UPPS	Adherence Mean (SD)	Non-Adherence Mean (SD)	*F*	η^2^
Lack of Premeditation	21.46 (6.53)	20.92 (5.86)	0.499 (ns)	0.008
Negative Urgency	32.11 (7.79)	33.56 (6.70)	160.437 ***	0.201
Lack of Perseverance	19.61 (6.49)	19.64 (8.05)	10,097 (ns)	0.016
Sensation Seeking	33.54 (7.82)	35.03 (8.23)	90.593 ***	0.128
Positive Urgency	32.64 (10.22)	33.81 (9.66)	100.627 ***	0.140

*Note*: *** *p* < 0.001; ns = not significant. η^2^ = The proportion of the total variance in the dependent variables. UPPS = Impulsivity Scale.

**Table 3 ijerph-18-08300-t003:** Mean, standard deviation, and significance level of differences Obsessive Belief Questionnaire (OBQ-44) between the groups.

OBQ-44	Adherent Mean (SD)	Non-Adherent Mean (SD)	*F*	η^2^
Responsibility/Inhibition	68.32 (15.20)	72.74 (16.96)	40.761 *	0.068
Perfectionism/Uncertainty	70.30 (15.84)	74.74 (17.96)	50.317 **	0.075
Importance/Control of thoughts	37.95 (12.32)	42.33 (14.74)	40.138 *	0.059

*Note*: ** *p* < 0.01; * *p* < 0.05; ns = not significant. η^2^ = The proportion of the total variance in the dependent variables. Obsessive Belief Questionnaire = OBQ-44.

## Data Availability

R codes and data are available from the authors on request.

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
