# Peer review of "Impulsivity and Compulsivity and Their Relationship with Non-Adherence to Treatment in the Prison Population"

_ijerph, 2021, doi:10.3390/ijerph18168300_

Round 1
Reviewer 1 Report
This study is the first investigation to analyze the role of impulsivity and compulsivity concerning adherence to treatment in the inmates.
The theme is important. Definition of the study variables are clearly defined.
The application of the study results could be elaborated more clearly.
Author Response
Reviewer 1:
This study is the first investigation to analyze the role of impulsivity and compulsivity
concerning adherence to treatment in the inmates.
Thank you very much for your positive comments
The theme is important. Definition of the study variables are clearly defined.
We really appreciate your comments
The application of the study results could be elaborated more clearly.
We have clarified these aspects by including. “The results of this research lead four ways to
clinical applications. Firstly, to design adequate frontal lobe dysfunction intervention. Secondly,
to encourage activities necessary for independent living at the prison or in the community.
Thirdly, to plan coping strategies that approach new challenges, tasks, and problems each day
and finally, fourthly, to performance of daily activities that achievement of the proposed goals
Reviewer 2 Report
According to the journal's guidelines, the abstract should follow the style of structured abstracts but without headings. Please, remove them.
Line 45 why "elderly age" is written in italic font?
Line119: why the authors chosen 55 years as an upper cut-off for age?
Line 128: SCL-90R – please write the proper name of the tool - The Symptom Checklist-90-R
Table 1: the fourth column has no heading. Please, add them
Information about obtaining the written consent was written twice.
Adherence/Non-adherence (in methodology) – please add information about the cut-off point for groups.
Please add (regard all of the using scales) whether the results are self-reported or assessed by a psychologist or other specialist. This information is essential to the interpretation of data results.
The authors should add the reference to the Declaration of Helsinki.
In the Statistical analysis section, please add the full name of the used test: likelihood-ratio test (chi-squared )
Table 1should be put in the results section, to my mind.
In the last column (table2), the correctness of the headline should be verified (as exact as table 3)
Author Response
Thank you very much for giving us the opportunity to resubmit our manuscript to IJERPH. We would also like to thank the reviewers for their positive comments and their clear feedback.
In this revised version we have made the required changes to the original manuscript, in order to address these comments. We outline these changes in an itemized fashion below. We hope the revisions are satisfactory and the manuscript is now suitable for publication in of IJERPH.
Reviewer 2:
According to the journal's guidelines, the abstract should follow the style of structured
abstracts but without headings. Please, remove them.
Done
Line 45 why "elderly age" is written in italic font?
Sorry about the mistake. The error has been corrected.
Line119: why the authors chosen 55 years as an upper cut-off for age?
We have chosen 55 years as an upper cut-off for age in order to prevent mental health issues
and neurocognitive deficits.
Curtis, A., Gooden, J. R., Cox, C. A., Harries, T., Peterson, V., Enticott, P. G., . . . Manning, V. (2021).
Neurocognitive functioning among people accessing an addiction neuropsychology clinic with and without
a history of offending behaviour. Psychiatry, Psychology and Law,
doi:http://dx.doi.org/10.1080/13218719.2021.1873204.
Kavanagh, L., Rowe, D., Hersch, J., Barnett, K. J., & Reznik, R. (2010). Neurocognitive deficits and
psychiatric disorders in a NSW prison population. International Journal of Law and Psychiatry, 33(1), 20-
26. doi:http://dx.doi.org/10.1016/j.ijlp.2009.10.004.
O'Rourke, C., Linden, M. A., Lohan, M., & Bates-Gaston, J. (2016). Traumatic brain injury and co-occurring
problems in prison populations: A systematic review. Brain Injury, 30(7), 839-854.
doi:http://dx.doi.org/10.3109/02699052.2016.1146967.
Line 128: SCL-90R – please write the proper name of the tool - The Symptom Checklist-90-R
Sorry about the mistake. The error has been corrected.
Table 1: the fourth column has no heading. Please, add them
Sorry about the mistake. The mistake has been added.
Information about obtaining the written consent was written twice.
Sorry about consent was written twice. We have deleted one written of them.
Adherence/Non-adherence (in methodology) – please add information about the cut-off point
for groups.
In procedure section we explained that: the Adherence versus Non-Adherence elements were:
Compliance with out-of-session activities after the intervention session (known as homework);
completion of self-reports; maintenance of abstinence from alcohol and drug use (tobacco was
not included); attendance at therapeutic sessions, attendance at workshops; participation in
leisure activities programmed by the prison in the Sociocultural Center (e.g., cinemas, concerts,
and conferences); daily physical activity; maintenance of personal cleanliness, personal area
(cell) and common areas; eating the food prepared in the Prison Center (because it is balanced
and healthy) and avoiding buying and eating junk food from the commissary. This information
was transformed into a dichotomous variable (Adherence, Non-Adherence).
Please add (regard all of the using scales) whether the results are self-reported or assessed by
a psychologist or other specialist. This information is essential to the interpretation of data
results.
We have added in the Methods that the participants filled as pen-and paper by the participants
themselves. This process was supervised by psychologists. All patients were informed about the
study course and methods, and about the possibility of withdrawing from the study at any time.
All patients provided their written informed consent to participate in the anonymous survey.
The authors should add the reference to the Declaration of Helsinki.
Sorry about the mistake. The mistake has been added.
In the Statistical analysis section, please add the full name of the used test: likelihood-ratio
test (chi-squared ).
The mistake has been corrected
Table 1should be put in the results section, to my mind.
We think table 1 should be in the participants' section because it describes the characteristics
sociodemographic, so it would be unnecessarily be put in the results section.
In the last column (table2), the correctness of the headline should be verified (as exact as table
3)
Done
Also we have revised all your comments, such as: revise our manuscript according to the
referees’ comments, reconfirm the author list and the corresponding affiliations and correct
spelling of authors' name.